# Utility of the Ribosomal Gene 18S rRNA in the Classification of the Main House Dust Mites Involved in Hypersensitivity

**DOI:** 10.3390/ijms262110308

**Published:** 2025-10-23

**Authors:** Antonio García-Dumpierrez, David Rodriguez Gil, M. Dolores Gallego Segovia, Javier Alcover, Montserrat Martínez-Gomariz, Aida Gómez, Ricardo Palacios

**Affiliations:** 1Hospital Universitario Dr. Negrín, 35010 Las Palmas de Gran Canaria, Spain; antoniodumpierrez0@gmail.com; 2Laboratorios DIATER I+D, 28918 Leganés, Spain; d.rodriguez@diater.com (D.R.G.); m.martinez@diater.com (M.M.-G.); a.gomez@diater.com (A.G.); r.palacios@diater.com (R.P.); 3Servicio de Alergia, Complejo Hospitalario Insular Materno Infantil de Gran Canaria, 35016 Las Palmas de Gran Canaria, Spain; magase1@yahoo.es

**Keywords:** allergy, mites, identification, PCR, sequences

## Abstract

Between 1% and 2% of the world’s population is sensitised to mites. Aetiological diagnosis is key to the management of allergic patients. However, methods based solely on morphological criteria are ambiguous in many cases. Polymerase chain reaction of ribosomal genes represents a valuable complementary approach. The 5 most representative species (*Dermatophagoides pteronyssinus*, *Dermatophagoides farinae*, *Tyrophagus putrescentiae*, *Blomia tropicalis* and *Lepidoglyphus destructor*) were selected as sources of allergens. They were first identified morphologically and the 18S rRNA gene sequences were obtained from the GenBank database. Alignment of the nucleotide sequences of the 18S rRNA ribosomal gene enabled the identification of the conserved and divergent regions in all of them. The alignment allowed the design of a pair of oligonucleotides in conserved regions of the gene, to amplify the sequence of interest in each of the species. We performed genomic DNA extraction, quantification and purity. PCR, using oligonucleotides designed to amplify the 18S sequence fragment of interest, showed the exact size for each species. Amplification, efficiency curves and melting points resulting from the amplification of the 18S amplicon of the five species were obtained. The oligonucleotides designed for real-time PCR studies, allow species identification by amplifying the specific fragment of each species using real-time PCR.

## 1. Introduction

The relationship between house dust and asthma was first proposed over 300 years ago by Sir John Floyer [1]. The presence of mites in the dust was discovered half a century ago by Reindert Voorhorst and the married couple, F.T. Spieksma and M.I. Spieksma-Boezemanin [2], in The Netherlands in a sample of house dust. To date, more than 45,000 species of mites have been described, with house dust mites being the most prevalent source of sensitisation of the so-called indoor allergens and the second most prevalent of allergens after pollens, being associated with the most common allergic diseases, such as asthma, rhinitis and atopic dermatitis [3,4]. It is estimated that between 1% and 2% of the world’s population is sensitised to mites, representing more than 130 million people could be allergic to them [5]. The domestic mite fauna involved in IgE-mediated hypersensitivity reactions belong to the class Arachnida, subclass Acari, suborder Astigmata, family Pyroglyphidae, which includes the genus *Dermatophagoides* and *Euroglyphus*, the family Echymyopodidae, which includes the genus *Blomia*, the family Acaridae, which includes the genus *Tyrophagus* and *Acarus* and the family Glycyphagidae, which includes the genus *Glycyphagus*, *Lepidoglyphus* and *Gohieria* [6]. The most representative species and with a high number of homologous allergens are *Dermatophagoides pteronyssinus* (DPT), *Dermatophagoides farinae* (DF), * Lepidoglyphus destructor* (LD), *Tyrophagus putrescentiae* (TP) and *Blomia tropicalis* (BT) [7,8].

Aetiological diagnosis is key to the management of allergic patients. Understanding both the source of the allergen and the allergen sensitisation profile is essential in order to establish appropriate avoidance measures and to restore the altered immunity of the allergic patient through allergen immunotherapy. Currently approximately 40 groups of mite allergens are recognised by the World Health Organization and International Union of Immunological Societies (WHO/IUIS), 37 of which belong to *D. farinae* and 34 to *D. pteronyssinus*. There is a clear serodominance for groups 1 and 2 of the genus *Dermatophagoides* and for the allergen Der p 23 [9,10]. Dust mite allergens are found mainly in their faeces and exhibit phylogenetic conservation with high structural homology, which allows them to be grouped and explains the phenomenon of cross-reactivity.

Traditional methods based on morphological criteria and exoskeleton characteristics [11,12] to distinguish between different species are often ambiguous in many cases, although they continue to be recommended in the Guidelines of health agencies [13]. However, the molecular biology techniques such as polymerase chain reaction (PCR) amplification targeting ribosomal genes (18S, 5.8S, and 28S), which contain very conserved areas, are currently very advanced and represent a valuable additional option in terms of replacing the classic morphological ones. Sequencing of the 18S ribosomal ribonucleic acid (rRNA) gene from several mite species and alignment of these sequences with those in the GenBank has been used in the identification of different taxa and has demonstrated the utility of these genetic markers in taxonomic and phylogenetic studies [14,15,16].

This study proposes a real-time PCR method targeting variable regions of the 18S rRNA gene to identify five dust mite species (*D. pteronyssinus*, *D. farinae*, *L. destructor*, *T. putrescentiae* and *B. tropicalis*). Our aim is to develop species-specific primers using melting curve analysis and to demonstrate the usefulness of this method for identifying mite populations as well as quality control in allergen extract production.

## 2. Results

### 2.1. Amplification of the 18S rRNA Gene of the Different Species

The genomic DNA extracted from the five species had very high levels of purity (Appendix A) shows the values for the five species. Regarding the 260/280 absorbance (A) ratio, we found 2 samples at optimal values (1.8–2) and 3 (*D. farinae*, *T. putrescentiae* and *B. tropicalis*) 2 above (standard deviation 0.12), indicating the presence of RNA. The ratio 260/230 was optimal (1.8–2.2) for 3 samples and low (1.7 and 1.6) for *B. tropicalis* and *L. destructor* (standard deviation 0.25).

Figure 1 shows the agarose gels of the fragments resulting from the amplification of the 18S rRNA gene. The first lane corresponds to the 100 base pairs (bp) marker and the second lane to the amplified DNA from the mite sample, and we can tell that the PCR result is correct since the sizes of the bands coincide.

### 2.2. 18S Amplification with Specific Oligonucleotides by Real-Time PCR

Figure 2 shows the amplification curves (left), efficiency curves (centre) and melting points (right) obtained from the amplification of the 18S amplicon of *D. pteronyssinus* as a representative example. Data for all five species studied are shown in Appendix A. To generate the efficiency curves a tenfold serial dilution of DNA was used. The melting points obtained were 74.5, 72.5, 76.5, 74.0 and 71.5 °C, respectively. Both the amplification curves and the correlation between dilutions were correct, as the efficiency curves fit the expected parameters. The melting curves obtained for each species showed a single peak, demonstrating the specificity of the amplification.

### 2.3. Specificity of Oligonucleotides

Figure 3 illustrates the specificity of the oligonucleotides, using *D. pteronyssinus* as a representative example, both with its specific primer pair and with those designed for the rest of the species. Each reaction was performed in triplicate. The data of the five species studied are in Appendix A.

For *D. pteronyssinus* (Figure 3, DPT curve in green; curves for other species in magenta), amplification with its specific primers (green) occurred at a cycle threshold (Ct) of 19.5 whereas amplification with primers specific to other species (magenta) occurred at Ct values above 30, therefore non-specific. Its melting curve (Figure 3) showed the amplification curve corresponding to the specific oligonucleotides (green), with a peak at 74.5 °C. Although amplifications with heterologous oligonucleotides show a melting curve with a maximum at that temperature, they also show additional dissociation peaks at other temperatures. The green horizontal line indicates the fluorescence threshold and represents the critical point at which the Ct exceeds a potential background noise.

For *D. farinae*, amplification with specific oligonucleotides (Appendix A) occurred at a Ct of 21.0. Amplification with nonspecific oligonucleotides occurred at Cts greater than 30 °C, it was therefore considered outside the detection range. The melting curve (Appendix A) shows that the dissociation point corresponding to the specific oligonucleotides (magenta) occurred at 72.5 °C. Although amplifications with heterologous oligonucleotides also showed a maximum at that temperature, the overall curve shape was different and additional melting peaks were observed at other temperatures. The magenta horizontal line represents the fluorescence threshold and is the point where the Ct is considered to exceed a possible background noise.

For *T. putrescentiae* (Appendix A) green line shows the amplification corresponding to the specific oligonucleotides and in other colours, the curves obtained with the heterologous oligonucleotides corresponding to the other species. Amplification with specific oligonucleotides occurred at Cts of 18.7, whereas amplification with heterologous oligonucleotides occurred at Cts values above 30, thus falling outside the detection range. The melting curve (Appendix A) shows that the dissociation point corresponding to the specific oligonucleotides (green) occurred at a temperature of 76.5 °C, while the heterologous oligonucleotides give rise to curves with different melting temperatures. The green horizontal line (right-hand image) indicates the fluorescence threshold and represents the critical point at which the Ct exceeds a possible background noise.

For *B. tropicalis*, Appendix A shows the amplification curves obtained using both specific and heterologous oligonucleotide pairs. The amplification curves obtained with oligonucleotides corresponding to *B. tropicalis* are shown in blue, and the curves obtained with oligonucleotides corresponding to the other species are shown in orange. Amplification with specific oligonucleotides occurred at Cts of 21.1 and amplification with nonspecific oligonucleotides, at Cts above 30, so it is considered outside the detection range. In the melting curve (Appendix A) the specific dissociation peak corresponding to the specific oligonucleotides (blue) occurred at a temperature of 74.0 °C. Heterologous oligonucleotides give curves with different temperatures, 67–68 °C, indicating the amplification of unspecific fragments with different lengths. The green horizontal line indicates the fluorescence threshold, and is the point where the Ct is considered to exceed a possible background noise.

For *L. destructor*, Appendix A shows the amplification curves obtained with oligonucleotides specific for the genomic DNA of *L. destructor* (violet), and the curves obtained with the oligonucleotides corresponding to the other species (orange). As can be seen, amplification with specific oligonucleotides occurred at Cts of 15.5. In contrast, amplification with nonspecific oligonucleotides occurred at Cts well above 30, so it is considered outside the detection range. The melting curve (Appendix A) shows the specific peak of the specific oligonucleotides (violet) at a temperature of 71.5 °C. Amplifications with heterologous oligonucleotides show peaks at different temperatures. The green horizontal line indicates the fluorescence threshold and is the point where the Ct is considered to exceed a possible background noise.

Table 1 (below) shows a summary presenting for each species: oligonucleotide sequence, melting Temperature (Tm), PCR efficiency, R^2^ value, and the Ct value for self vs. non-self-amplification.

### 2.4. Oligonucleotides Species-Specificity (Results for D. pteronyssinus)

Figure 3 (left) shows the amplification curves obtained by testing *D. pteronyssinus* genomic DNA from a pure sample with all oligonucleotide pairs. The amplification curves obtained with the *D. pteronyssinus*-specific oligonucleotides are shown in green, and the curves obtained with the oligonucleotides corresponding to the other species are shown in magenta. Amplification with the specific oligonucleotides occurs at a Ct of 19.5. In contrast, the Ct values for amplification with the nonspecific oligonucleotides are above 30, so it is considered outside the detection range.

The corresponding melting curve (Figure 3, right) shows the amplification curve corresponding to the specific oligonucleotides (green), with a peak at a temperature of 74.5 °C. Although amplifications with heterologous oligonucleotides show a melting curve with a maximum at that temperature, they also present other dissociation peaks at other temperatures.

In the genomic DNA sample obtained from the extraction with culture medium (Figure 3), we obtained very similar results, since the oligonucleotides designed for the other species also amplify at Cts above 30, and in addition their melting curves present more than one peak, allowing, even, a better discrimination.

### 2.5. Specificity of Oligonucleotides in Artificially Contaminated Cultures

Samples of *D. pteronyssinus* (in culture, 230 mg) were analysed with contaminant X (*B. tropicalis*) in the following quantities: Vial 1: 1 mite; Vial 2: 5 mites; Vial 3: 15 mites; Vial 4: 20 mites and Vial 5: 25 mites.

Once the extraction was carried out, the DNA quantification gave the following results: Vial 1 (4.24 µg/mL; A260/A280 = 1.57; A260/A230 = 0.53. Vial 2 (4.79 µg/mL; A260/A280 = 1.64; A260/A230 = 0.52. Vial 3 (5.07 µg/mL; A260/A280 = 1.70; A260/A230 = 0.54. Vial 4 (3.84 µg/mL; A260/A280 = 1.61; A260/A230 = 0.47. Vial 5 (3.52 µg/mL; A260/A280 = 1.60; A260/A230 = 0.47.

Due to the excess medium, which is not completely removed after DNA extraction, the absorbance values at 260/280 and 260/230 indicate that the extracted DNA is not completely pure. Nevertheless, we continued with the quantitative PCR (qPCR) analysis to observe whether these medium residues affect amplification.

The qPCR results for each sample vial are detailed below. The graphs on the left correspond to the amplification curves, and those on the right correspond to the melting curve. Fifty nanograms of genomic DNA were analysed from each sample.

Figure 4a–e shows amplification curve and melting peak of a for each vial *D. pteronyssinus* (in culture, 230 mg) contaminated with 1, 5, 15, 20 and 25 mites each one of an unknown specie.

Figure 4: Amplification curve and melting peak of a for each vial *D. pteronyssinus* (in culture, 230 mg) contaminated with 1, 5, 15, 20 and 25 mites (Figure 4a–e) of an unknown species.

Analysis of these results confirmed that the contaminating species was *B. tropicalis*, since it is the only species that amplifies with Ct values below 30 and the melting curve peak appears at 73.5 °C. The sensitivity of the assay was 15 mites, since this is the limit at which the difference in amplification is significant enough to distinguish it.

## 3. Discussion

In clinical allergy practice, allergenic extracts are used, which are obtained from cultures of mite species that inhabit house dust or dust derived from the storage of various plant or animal foods. Contamination of house or warehouse dust mite cultures, or misclassifications of the species from which allergenic extracts or allergens used in aetiological diagnosis are produced or purified, may compromise clinical outcomes in patients undergoing allergen-specific immunotherapy, both in terms of efficacy, as the allergen is not the one to which the patient is sensitised, and safety of use, by increasing the risk of sensitisation to other allergens not present in the primary source of sensitisation.

The primary objective of this study was to facilitate the identification of mite species present in patients’ homes. The decision regarding the appropriateness of allergen-specific immunotherapy and its exact composition is the sole responsibility of the treating allergist and should be guided by current European and American guidelines.

This study presents a real-time PCR-based method to distinguish among the species of mites most commonly involved in the sensitization of house dust mite (HDM) allergic individuals. The rationale behind this is that different species of mites contain similar but not identical allergens, and that the species discrimination could help the practitioner with the diagnosis.

In this work, we present a simple method, based on genetic markers, to ensure control of the specific culture of the most significant species involved in house dust mite allergy and in house dust samples. We used different tools, such as the design of specific oligonucleotides that amplify fragments exclusive to each of the species.

The oligonucleotides designed for real-time PCR studies, the yield and quality of the extraction of genomic DNA from each mite species, the cloning of the fragments of interest, the purification of plasmids, and the amplification and sequencing of the exclusive fragments have all proven to be adequate and of high quality, showing specific variants in some of the mite species.

Regarding sensitivity, the system was able to detect as few as 15 individual mites, since this is the limit at which the difference in amplification is significant enough to distinguish it.

The use of these nucleotides allows species identification by amplifying the specific fragment of each species using real-time PCR. Once implemented, it will be a simple, fast, and efficient method. It could be useful for identifying allergy triggers in homes, hospitals, and schools, providing a fast and practical tool to support current diagnostic procedures.

## 4. Materials and Methods

Five of the most representative species of house and storage dust mites were selected as sources of allergens. They were identified morphologically from certified cultures (Allergen Servilab, La Horcajada, Avila, Spain) and their 18S rRNA gene sequences were obtained from the GenBank database. The culture medium consisted of equal parts of plant feed and hydrolysed yeast. Each culture contained approximately 60–70 adult mites per milligram, as well as other unquantified larval stages. This gene is highly specific and is therefore used in species identification studies and allows the design of specific oligonucleotides for each sequence and to identify each of the species by quantitative reverse transcription PCR (RT-qPCR): *D. pteronyssinus* (KC215344.1), hereinafter DPT; *D. farinae* (KC215333.1), hereinafter DF; *T. putrescentiae* (KC215293.1), hereinafter TP; *B. tropicalis* (KC215356.1), hereinafter BT; and *L. destructor* (KC215384.1), hereinafter LD.

Alignment of the nucleotide sequences of the 18S rRNA ribosomal gene of the five species allowed the identification of conserved and divergent regions in all of them (Appendix A). The alignment allowed the design of a pair of oligonucleotides in conserved regions of the 18S rRNA ribosomal gene, to amplify the sequence of interest in each of the species. The sequences of the oligonucleotide primers designed were Forward: 5′-AGTGGTGGATCACTCGGC and Reverse: 5′-TCTCGCTTGATCGAGGTC.

Although these primers are conserved across all species, the fragment included between them is highly variable and species-specific, enabling the design of additional primers to unequivocally distinguish each mite species. The sequences of the oligonucleotides designed to specifically identify each of the species are shown in the following table (Table 2):

We performed genomic DNA extraction using the DNA tissue isolation kit (NZYTech, Lisboa, Portugal) from 10 mg of each of the mite samples and we quantified them by spectroscopy, measuring the absorbance at 260 nm and the ratios 260/280 and 260/230 to estimate the quality of the extraction.

PCR, using oligonucleotides designed to amplify the 18S rRNA ribosomal gene sequence fragment of interest, showed the exact size for each species: DPT: 397 bp; DF: 403 bp; TP: 554 bp; BT: 390 bp; and LD: 540 bp.

The general conditions for carrying out the PCR were 1 µL of 10 µM sense and antisense oligonucleotides, 25 µL PCR MasterMix (ThermoFisher, Waltham, MA, USA), 19 µL H_2_O (sterile), 4 µL of undiluted extracted DNA, using the following steps: (1) Initial denaturation (94 °C-3 min); (2) 36 PCR cycles (94 °C-3 s; 48 °C-1.30 min; 72 °C-10 min); (3) Final extension (72 °C-10 min).

Once the size of the amplified fragment was confirmed and cloned into a pSTBlue-1 expression vector (Novagen/EMD Millipore, Billerica, MA, USA), it was sequenced, maintaining the plasmid containing the sequenced fragment in permanent culture as a reference.

The specificity of the oligonucleotides of the five species was verified by qPCR, establishing the Ct values (number of cycles required for the fluorescent signal to exceed the background level). Ct values are inversely proportional to the amount of amplified DNA from the sample [17].

Efficiency curves were generated for each pair of oligonucleotides tested with genomic DNA from the corresponding species, using a dilution bank to evaluate its efficiency. Genomic DNA was diluted to an initial concentration of 5 ng/μL with three serial dilutions and final analysis by real-time PCR with 10 ng, 1 ng, 0.1 ng and 0.01 ng of genomic DNA. In the case of DPT, 3 ng, 0.3 ng, 0.03 ng and 0.003 ng were used.

At the end of the amplification cycles, we carried out a dissociation (melting) curve analysis to identify the dissociation temperature corresponding to the amplified fragment (or amplicon) for each pair of oligonucleotides [18]. The temperature of this curve also allows each species to be identified. The dissociation temperature, which corresponds to the temperature at which denaturation of the DNA strands occurs, is specific for each amplicon, such that different melting temperatures identify different amplicons [19].

The qPCR program consisted of an initial 3 min at 95 °C, followed by 45 cycles of 10 s at 95 °C and one minute at 55 °C, ending with a melting curve, increasing the temperature from 55 °C to 95 °C.

Once specificity was established for each oligonucleotide to amplify a single sequence in each species, we went on to demonstrate that each pair of selected oligonucleotides was species-specific. Genomic DNA of each species was tested with oligonucleotides designed for the other species. Amplification only occurred when the species-specific oligonucleotides were used (specific oligonucleotides versus nonspecific or heterologous oligonucleotides).

We demonstrated the specificity of each pair of oligonucleotides selected by species using 10 ng of genomic DNA from each species with the oligonucleotides designed for the five species and confirmed that amplification only occurred when using the oligonucleotides of the species for which they were selected (positive control).

We determined the level of detection of possible contaminating mite species by means of genomic DNA analysis of the different species. For each species, a 50 ng/µL dilution of genomic DNA from one species was prepared and then contaminated with 0.1 ng/µL of genomic DNA from each of the other species. Finally, we analysed 5 ng of each main sample pair together with the contaminating mite sample, the specific oligonucleotides of the main sample and the corresponding oligonucleotides of the contaminating species. In this case, genomic DNA extraction was performed manually by precipitating the DNA with isopropanol, for which a lysis buffer was prepared (Tris-HCl, EDTA, SDS and proteinase K). After homogenising the samples, they were incubated at 50 °C for 16 h and then gently centrifuged to discard the precipitate. We added 1.6 volumes of isopropanol to the supernatant and repeated the centrifugation, after which we discarded the supernatant and washed the precipitate containing the genomic DNA twice with 70% ethanol, dissolving it in sterile water and storing the samples at −20 °C.

Based on the results obtained across all analyses (we will show the results obtained for *D. pteronyssinus*), we establish that: there is contamination of species B in the culture of species A when, when using the pair of oligonucleotides from B with genomic DNA from A, an amplification with a Cts value below 30 is obtained and the temperature of the peak of the melting curve corresponds to that described for amplification with the oligonucleotides from the contaminating species.

The sample with its specific oligonucleotide pair was included in the analysis as a positive control. Three technical replicates were made for each sample, using 10 ng of genomic DNA.

An additional objective was to evaluate the method using samples contaminated with increasing numbers of mites, without knowing the contaminating species. The contaminating species will be referred to as contaminant X. The results obtained with *D. pteronyssinus* are detailed. Samples of *D. pteronyssinus* (in culture, 230 mg) were analyzed with contaminant X in the following quantities: Vial 1: 1 mite; Vial 2: 5 mites; Vial 3: 15 mites; Vial 4: 20 mites and Vial 5: 25 mites.

In this case, considering that it was necessary to process the entire sample to avoid losing contaminating mites, it was decided to opt for manual genomic DNA extraction (DNA precipitation by isopropanol), instead of using the commercial kit. Briefly, a lysis buffer was prepared (Tris-HCl, EDTA, SDS, and proteinase K). The samples were homogenized and incubated at 50 °C for 16 h. The samples were then centrifuged to remove the precipitate. 1.6 volumes of isopropanol were added to the supernatant and centrifuged again, retaining the precipitate containing the genomic DNA. The precipitate was washed twice with 70% ethanol, dissolved in sterile water and stored at −20 °C.

## Figures and Tables

**Figure 1 ijms-26-10308-f001:**
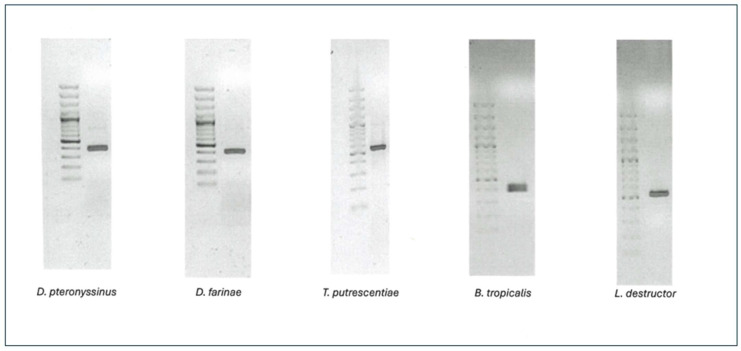
Amplification of the 18S rRNA gene of the different species. On left side is indicated the molecular weight label.

**Figure 2 ijms-26-10308-f002:**
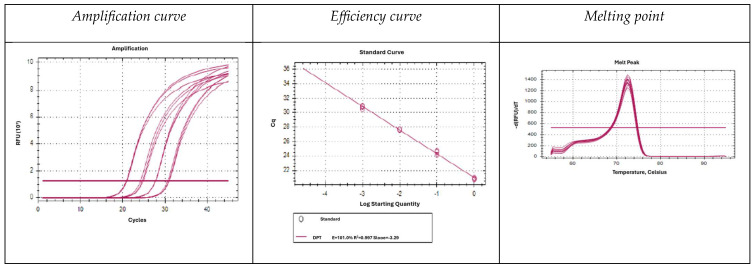
Amplification curves (**left**), efficiency curves (**centre**) and melting points (**right**) of *D. pteronyssinus*.

**Figure 3 ijms-26-10308-f003:**
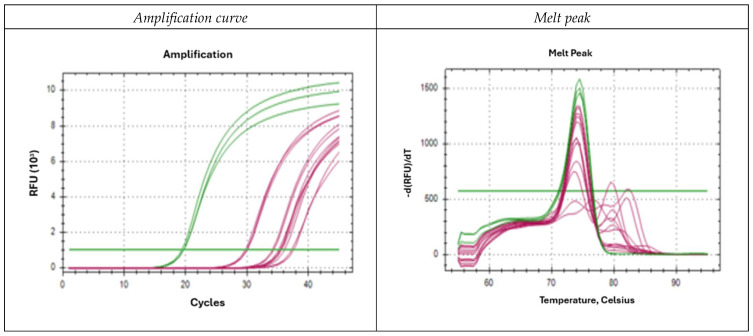
Amplification curves (**left**) and melting points (**right**) of *D. pteronyssinus* both with its specific primer pair and with those designed for the rest of the species.

**Figure 4 ijms-26-10308-f004:**
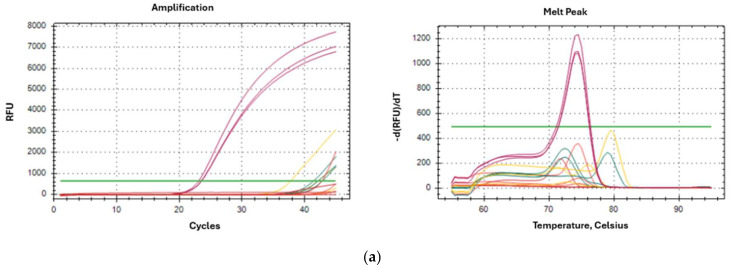
(**a**) DPT culture contaminated with 1 mite. (**b**) DPT culture contaminated with 2 mites. (**c**) DPT culture contaminated with 15 mites. (**d**) DPT culture contaminated with 20 mites. (**e**) DPT culture contaminated with 25 mites.

**Table 1 ijms-26-10308-t001:** Oligonucleotide sequence, melting Temperature (Tm), PCR efficiency, R^2^ value, and the Ct value for self vs. non-self-amplification for each species.

	Oligonucleotide Sequence Forward/Reverse	Tm °C	PCR Efficiency %	R^2^	Ct °C	Ct (Other) °C
DPT	5′-TGACTTTTGTGGTGAAGAAG-3′ 5′-CTCACAAGCGGTATTTAGCGT-3′	74.5	98.8	0.997	19.5	>30
DF	5′-ATTCATTTCGGTGTCGTGAG-3′ 5′-CATTAAAAAATCACCAAAGG-3′	72.5	101.0	0.997	21.0	>30
TP	5′-CTGCCCATACGAGCGTAGG-3′ 5′-TGAATCGAGTACCAGCAAAC-3′	76.5	95.8	0.999	18.7	>30
BT	5′-AAAGGAGACTTAAATAAGTTGTC-3′ 5′-GTTTTTCTCAATAAGTGAACA-3′	74.0	116.9	0.943	21.1	>30
LD	5′-TTTGAAATTTGGTGGTATTTG-3′ 5′-CAATATCAATTGTAATCACAA-3′	71.5	91.1		15.5	>30

**Table 2 ijms-26-10308-t002:** Sequences of the oligonucleotides designed.

Species	Forward	Reverse
*D. pteronyssinus*	5′-TGACTTTTGTGGTGAAGAAG-3′	5′-CTCACAAGCGGTATTTAGCGT-3′
*D. farinae*	5′-ATTCATTTCGGTGTCGTGAG-3′	5′-CATTAAAAAATCACCAAAGG-3′
*T. putrescentiae*	5′-CTGCCCATACGAGCGTAGG-3′	5′-TGAATCGAGTACCAGCAAAC-3′
*B. tropicalis*	5′-AAAGGAGACTTAAATAAGTTGTC-3′	5′-GTTTTTCTCAATAAGTGAACA-3′
*L. destructor*	5′-TTTGAAATTTGGTGGTATTTG-3′	5′-CAATATCAATTGTAATCACAA-3′

## Data Availability

The original contributions presented in this study are included in the article. Further inquiries can be directed to the corresponding author.

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
