# Peer review of "Utility of the Ribosomal Gene 18S rRNA in the Classification of the Main House Dust Mites Involved in Hypersensitivity"

_ijms, 2025, doi:10.3390/ijms262110308_

Round 1
Reviewer 1 Report
Comments and Suggestions for Authors
Dear authors,
I have carefully read your article and believe that it is necessary to include major revisions before publishing it. Below are the individual points for editing and comments.
- lines 54 and 55: numbers 37 and 34 should not be bold
- line 71: "2.1.1.-Genomic DNA..." Please remove the hyphen in the title (-Genomic), as well as in the rest of the text.
- lines 73 - 74: Unify the style of writing numbers (three vs 2 in the text "we found three samples at optimal values (1.8-2) and 2 (T. putrescentiae and B. tropicalis)" ).
- line 74: I would prefer a range or mean with a standard deviation for absorbance values rather than a vague "slightly above 2"
- Table 2:
- It is the first table listed, not the second. Correct the errors and verify the numbering of tables and figures throughout the text.
- You wrote the concentration units for each value separately, but for the number of bases for amplicons, "bp" is only in the table header (Size bp). Choose one style for both.
- It would be appropriate to state the standard deviations or the range of the obtained values. Or were all the numbers measured only once??
- Unify the number of decimal places: for Blomia tropicalis add "59.0".
- line 79: "Figure 1." without a dot and probably not in bold
- Figure 1: bands (amplicons) should not be drawn in red, but leave a clean picture, as in B. tropicalis
- lines 89 - 90: details for figure 2 ("from top to bottom: Dermatophagoides pteronyssinus, Dermatophagoides farinae, Tyrophagus putrescentiae, Blomia tropicalis and Lepidoglyphus destructor") up to the legend of this figure, here it is irrelevant
- lines 91-92: why were different concentrations used to determine the LOD?
- line 93: I think there should be a space between the number and the degree Celsius. Please correct it throughout the text.
- figure 2:
- In the header, there is only "melting curve" in italics - unify with amplification and efficiency curve
- The letter labels (a, b etc) are not visible - maybe it would be better to put the image on its own page so that it doesn't get messy. Also, the efficiency curve in the case of "d" extends beyond the field (above) and e/f cover the field dividing line. Align it.
- At point „f“ the curves are quite ugly. How do you explain that?
- Line 99: „Figure 2. amplification curves“ – use the capital „A“ in the „amplification“
- Lines 99 – 101: all listed species write in italics
- why is the "curve" column empty in the table under the pictures? Theoretically, the table is not needed; just move the sentence from lines 90-92 to the picture caption (or just to the methodology) and state the melt peak next to the melting curve - it will fit there. Moreover, the only thing that is not readable in the pictures is not stated in the table (calibration curve parameters).... So either add Eff, R2, slope to the table or rewrite it in the pictures so that it is readable, add Tm as well and delete the table. You stated the used DNA dilution series concentrations in the text, it is not necessary to repeat it there.
- line 111: in the table above (Fig. 2), Tm = 77 °C is given for DPT - why is 74.5 °C expected here?
- 3: Tm of target and non-target species overlap - how would you differentiate if there was a lower DNA concentration (and therefore a later Ct)? You do write that specific amplification curves are with Ct 19 and non-specific 30, but you do not comment on this at all.
- the markings a, b etc. are not visible at all; it would be a good idea to make the pictures larger so they are easier to read
- the legend of the figure is in bold here; in Fig. 2 it is not. Unify according to the instructions for authors
- Line 117: I would appreciate some average Ct values, it certainly didn't come out 21.0 every time.
- Lines 121-122: „the shape of the 121 curve was different and additional melting peaks were observed at other temperatures“ - I don't see that, it seems pretty much the same to me - maybe it would be good to choose a color other than yellow (it is hard to see)... Why don't you keep green = specific, magenta = non-specific species for all curves in Fig. 3?
- Lines 125, 135, 147: unify the style - somewhere the species is stated in the sentence (Tyrophagus putrescentiae (Figure 3e) in green shows), somewhere with a dot after the species (Blomia tropicalis. Figure 3g ) and somewhere in bold (Lepidoglyphus destructor. Figure (3i))
- Line 143: „Heterologous oligonucleotides give curves with different temperatures, 67-68ºC,“ - why don't you provide Tm values of non-specific peaks for other species?
- Discussion: deepen the discussion, include comparisons with other authors/previously published articles. This is more summary of results rather than their discussion.
- Lines 172 – 173: „We used different tools, such as the design of specific oligonucleotides, designing primers that amplify fragments exclusive to each of the species“ - That's the same thing twice (specific oligonucleotides and primers specific for the given mites), delete one of them.
- Lines 195 – 196: is it possible list the sequences in Table 1 uniformly with the other primers? If these primers give products of different sizes - wouldn't it be possible to distinguish individual species based on Tm even using these universal primers?
- Line 207: „18S“ – please use the full name of gene in whole text.
- Line 214: please delete „and End of the reaction (15ºC-∞)“. This is not necessary to state, and moreover, if it were really infinite, you would still be doing it today.
- Line 266 – Abbreviations: please write all species in italics.
Author Response
Comments 1:
- lines 54 and 55: numbers 37 and 34 should not be bold
Answer 1:
Already changed in the manuscript.
Comments 2:
- line 71: "2.1.1.-Genomic DNA..." Please remove the hyphen in the title (-Genomic), as well as in the rest of the text.
Answer 2:
Already changed in the manuscript.
Comments 3:
- lines 73 - 74: Unify the style of writing numbers (three vs 2 in the text "we found three samples at optimal values (1.8-2) and 2 (T. putrescentiae and B. tropicalis)" ).
Answer 3:
Already changed in the manuscript.
Comments 4:
- line 74: I would prefer a range or mean with a standard deviation for absorbance values rather than a vague "slightly above 2"
Answer 4:
Already calculated and changed in the manuscript.
We extracted 10 mg of each culture to obtain material for the various tests and fine-tune the method. We have corrected your observations and will also add the table to the supplementary material, with the corrected numbering (Table 1 supplementary).
Comments 5:
- Table 2:
- It is the first table listed, not the second. Correct the errors and verify the numbering of tables and figures throughout the text.
- It would be appropriate to state the standard deviations or the range of the obtained values. Or were all the numbers measured only once?? *
- Unify the number of decimal places: for Blomia tropicalis add "59.0".
Answer 5:
*Dolors: Es la misma respuesta que antes
+ Already changed in the manuscript.
Comments 6:
- line 79: "Figure 1." without a dot and probably not in bold
Answer 6:
Already changed in the manuscript.
Comments 7:
- Figure 1: bands (amplicons) should not be drawn in red, but leave a clean picture, as in B. tropicalis
Answer 7:
Already changed in the manuscript.
Comments 8:
- lines 89 - 90: details for figure 2 ("from top to bottom: Dermatophagoides pteronyssinus, Dermatophagoides farinae, Tyrophagus putrescentiae, Blomia tropicalis and Lepidoglyphus destructor") up to the legend of this figure, here it is irrelevant
Answer 8:
Already changed in the manuscript.
Vigilad esta figura, teneis seis graficos para 5 especies. La b y la c son lo mismo, está repetida: CORREGIDO
Comments 9:
- lines 91-92: why were different concentrations used to determine the LOD?
Answer 9:
This is determined by the quantity of DNA, and the efficiency of the oligonucleotides. Dermatophagoides pteronyssinus gave the lowest DNA concentration (54.9 ng/µl). These data are now in Supplementary material (Table 1s).
Comments 10:
- line 93: I think there should be a space between the number and the degree Celsius. Please correct it throughout the text.
Answer 10:
Already changed in the manuscript
Comments 11:
- figure 2:
- In the header, there is only "melting curve" in italics - unify with amplification and efficiency curve Already changed in the manuscript
- The letter labels (a, b etc) are not visible - maybe it would be better to put the image on its own page so that it doesn't get messy. Also, the efficiency curve in the case of "d" extends beyond the field (above) and e/f cover the field dividing line. Align it. Already changed in the manuscript
- At point „f“ the curves are quite ugly. How do you explain that?
Finding species-specific oligos isn't easy, and in this case, there aren't many options for selecting specific oligos that would give amplicons that are sufficiently distinct from those of other species. However, the dilution bank amplifies every three Ct perfectly, and the melting curve is perfect.
- Line 99: „Figure 2. amplification curves“ – use the capital „A“ in the „amplification“
- Lines 99 – 101: all listed species write in italics YA ESTABAN….¿?
- why is the "curve" column empty in the table under the pictures? Theoretically, the table is not needed; just move the sentence from lines 90-92 to the picture caption (or just to the methodology) and state the melt peak next to the melting curve - it will fit there. Moreover, the only thing that is not readable in the pictures is not stated in the table (calibration curve parameters).... So either add Eff, R2, slope to the table or rewrite it in the pictures so that it is readable, add Tm as well and delete the table. You stated the used DNA dilution series concentrations in the text, it is not necessary to repeat it there.
Answer 11:
Already changed in the manuscript.
Comments 12:
- line 111: in the table above (Fig. 2), Tm = 77 °C is given for DPT - why is 74.5 °C expected here?
Answer 12:
I think there is a problem explaining figure 3. An error has occurred - the correct temperature is 74.5 and it has been corrected in the text.
Comments 13:
- 3: Tm of target and non-target species overlap - how would you differentiate if there was a lower DNA concentration (and therefore a later Ct)? You do write that specific amplification curves are with Ct 19 and non-specific 30, but you do not comment on this at all.
- the markings a, b etc. are not visible at all; it would be a good idea to make the pictures larger so they are easier to read
- the legend of the figure is in bold here; in Fig. 2 it is not. Unify according to the instructions for authors
Answer 13:
Now the text was modified in the new version of the manuscript. It was indicated that a very low of genomic DNA from contaminants was used. This contaminant can be detected, although in some cases with more than 30 Ct. This high number of Cts is due to the low quantity of material used for this species
Comments 14:
- Line 117: I would appreciate some average Ct values, it certainly didn't come out 21.0 every time.
Answer 14:
In the new version of the manuscript the numbers were adjusted to decimals
Comments 15:
- Lines 121-122: „the shape of the 121 curve was different and additional melting peaks were observed at other temperatures“ - I don't see that, it seems pretty much the same to me - maybe it would be good to choose a color other than yellow (it is hard to see)... Why don't you keep green = specific, magenta = non-specific species for all curves in Fig. 3?
Answer 15:
Sorry. It is impossible to change the colours in the program. The unspecific amplicons are given different dissociation peaks as show in the image
Comments 16:
- Lines 125, 135, 147: unify the style - somewhere the species is stated in the sentence (Tyrophagus putrescentiae (Figure 3e) in green shows), somewhere with a dot after the species (Blomia tropicalis. Figure 3g ) and somewhere in bold (Lepidoglyphus destructor. Figure (3i))
Answer 16:
Already changed in the manuscript
Comments 17:
- Line 143: „Heterologous oligonucleotides give curves with different temperatures, 67-68ºC,“ - why don't you provide Tm values of non-specific peaks for other species?
Answer 17:
We obtained a temperature range, because we non-specifically amplified amplicons of different sizes.
Comments 18:
- Discussion: deepen the discussion, include comparisons with other authors/previously published articles. This is more summary of results rather than their discussion.
Answer 18:
Already included in the discussion:
Once implemented, it will be a simple, fast, and efficient method. It will be highly useful for identifying allergy triggers in homes, hospitals, and schools. It's a rapid way to make a reliable diagnosis.
Comments 19:
- Lines 172 – 173: „We used different tools, such as the design of specific oligonucleotides, designing primers that amplify fragments exclusive to each of the species“ - That's the same thing twice (specific oligonucleotides and primers specific for the given mites), delete one of them.
Answer 19:
Already changed in the manuscript
Comments 20:
- Lines 195 – 196: is it possible list the sequences in Table 1 uniformly with the other primers? If these primers give products of different sizes - wouldn't it be possible to distinguish individual species based on Tm even using these universal primers?
Answer 20:
We have added in supplementary material the alignment of the 5 species, where the oligonucleotides can be located (Figure 1s) and the difference between each amplicon can be seen.
Based on our experience, using universal primers would yield different amplifications for each species, but we wouldn't know which species we were amplifying. Since we intend to use the method for diagnostic purposes, it must allow us to determine which species is contaminating the site, which is why it is necessary to design specific oligonucleotides that also offer different sizes.
Comments 21:
- Line 207: „18S“ – please use the full name of gene in whole text.
Answer 21:
Already changed in the manuscript
Comments 22:
- Line 214: please delete „and End of the reaction (15ºC-∞)“. This is not necessary to state, and moreover, if it were really infinite, you would still be doing it today.
Answer 22:
Already changed in the manuscript
Comments 23:
- Line 266 – Abbreviations: please write all species in italics.
Answer 23:
Already changed in the manuscript
Reviewer 2 Report
Comments and Suggestions for Authors
The manuscript proposes a real-time PCR method targeting variable regions of the 18S rRNA gene to identify five dust mite species (Dermatophagoides pteronyssinus, D. farinae, Euroglyphus maynei, Tyrophagus putrescentiae, and Blomia tropicalis). The authors claim to have developed species-specific primers. They attempted to validate them using melting curve analysis and highlight the usefulness of the method for quality control in the production of allergen extracts.
However, the presented study has a number of significant shortcomings:
- The 18S rRNA gene is highly conserved across arthropods and lacks sufficient interspecies variability for reliable species discrimination. Recent studies have established 28S rRNA (particularly divergent domains D5, D6, and D8) as superior barcodes for mites due to higher sequencing success rates (67–78%) and greater discriminatory power. The authors neither justify their choice of 18S rRNA nor cite literature supporting its utility over 28S rRNA or mitochondrial genes.
- Dust mite allergens exhibit cross-reactivity with shellfish and cockroaches. The manuscript does not address how genetic similarities in 18S rRNA might parallel allergenic cross-reactivity, limiting clinical relevance.
- No negative controls (e.g., non-target mites or environmental DNA) were tested to confirm primer specificity.
- Two species ( putrescentiae and B. tropicalis) had suboptimal 260/230 ratios (1.6–1.7), indicating contaminating polysaccharides/phenols that could inhibit PCR. The authors dismiss this without validation.
- Dilutions used in efficiency curves (e.g., 0.003 ng for pteronyssinus) lack biological context. Detection limits in complex matrices (e.g., dust samples) were not evaluated.
- The study fails to compare its method against morphological identification or established molecular markers (e.g., cox1). Without this, claims of accuracy are unsubstantiated.
- The authors assert that species misidentification compromises allergen immunotherapy but provide no data linking genetic identification to allergenic potency. Prior work shows that microbiome composition (e.g., Enterobacter in farinae) directly influences allergenicity , yet this is ignored.
- The study does not quantify group 1/2 allergens or correlate genetic markers with allergen expression, undermining its utility for extract standardization.
- Figure 1: Gel images lack molecular weight markers for all lanes.
- Table 2: Absent standard deviations for DNA concentration/purity metrics.
- No rationale for targeting 18S variable regions over established barcoding regions (e.g., ITS).
The manuscript describes a clinically unsubstantiated approach to tick identification. The choice of 18S rRNA is unclear, validation is insufficient, and the clinical relevance is not definitive. Given these shortcomings, the article is not suitable for publication in IJMS.
Author Response
Comments 1:
- The 18S rRNA gene is highly conserved across arthropods and lacks sufficient interspecies variability for reliable species discrimination. Recent studies have established 28S rRNA (particularly divergent domains D5, D6, and D8) as superior barcodes for mites due to higher sequencing success rates (67–78%) and greater discriminatory power. The authors neither justify their choice of 18S rRNA nor cite literature supporting its utility over 28S rRNA or mitochondrial genes.
Answer 1:
All the information available on different species of mites and the work previously carried out in our laboratories (Diater) led us to start with this species, but we do not rule out addressing other sequences that may be useful in the diagnosis of allergies.
Comments 2:
- Dust mite allergens exhibit cross-reactivity with shellfish and cockroaches. The manuscript does not address how genetic similarities in 18S rRNA might parallel allergenic cross-reactivity, limiting clinical relevance.
Answer 2:
The 18S rRNA gene is highly conserved across mites, shellfish and cockroaches. Anyway the predominant inhalant way of mite’s sensitization above cockroaches does not impact the objectives of this study. Regarding shellfish, food allergens are well known and its presence in processed foods must be warned.
Comments 3:
- No negative controls (e.g., non-target mites or environmental DNA) were tested to confirm primer specificity.
Answer 3:
The new experiments (sections 2.3 and 2.4) added represent new controls to the specificity of the primers.
Comments 4:
- Two species (putrescentiae and tropicalis) had suboptimal 260/230 ratios (1.6–1.7), indicating contaminating polysaccharides/phenols that could inhibit PCR. The authors dismiss this without validation.
Answer 4:
Table 2 has been added to Supplemental material.
Amplification with both common and specific primers occurred without problems.
Comments 5:
- Dilutions used in efficiency curves (e.g., 0.003 ng for pteronyssinus) lack biological context. Detection limits in complex matrices (e.g., dust samples) were not evaluated.
Answer 5:
The dilutions used are determined by the characteristics of the oligonucleotides. The detection of such small quantities is due to the difficulty of obtaining samples with a high number of mites. Regarding the complexity of dust samples, this is further compounded by the use of mites from laboratory cultures, where, in addition to the mites, there are quantities of culture medium consisting of plant feed and hydrolysed yeast extract.
Comments 6:
- The study fails to compare its method against morphological identification or established molecular markers (e.g., cox1). Without this, claims of accuracy are unsubstantiated.
Answer 6:
In our study, we used pure, properly identified laboratory mite cultures. Their identification and purity are unequivocal.
If needed, we can add microscope photos of the species used in supplementary material.
Comments 7:
- The authors assert that species misidentification compromises allergen immunotherapy but provide no data linking genetic identification to allergenic potency. Prior work shows that microbiome composition (e.g., Enterobacter in farinae) directly influences allergenicity , yet this is ignored.
Answer 7:
Obviously, there can be many other factors that cause allergies. But this is a first step toward identifying them. For now, being able to identify the different mites that affect the population differently is important, and future studies of the microbiota of these mites will surely help us understand allergies a little better.
Comments 8:
- The study does not quantify group 1/2 allergens or correlate genetic markers with allergen expression, undermining its utility for extract standardization.
Answer 8:
The objective of the study is to identify the species present in the patients' homes, not to characterize the extracts used for diagnosis or treatment that are prepared from pure mite cultures in the laboratory.
Comments 9:
- Figure 1: Gel images lack molecular weight markers for all lanes.
Answer 9:
Already changed in the manuscript
Comments 10:
- Table 2: Absent standard deviations for DNA concentration/purity metrics.
Answer 10:
We have added standard deviations in the manuscript. In addition, we will also add the table to the supplementary material, with the corrected numbering (Table 1 supplementary).
Comments 11:
- No rationale for targeting 18S variable regions over established barcoding regions (e.g., ITS).
Answer 11:
All the information available on different species of mites and the work previously carried out in our laboratories (Diater) led us to start with this species, but we do not rule out addressing other sequences that may be useful in the diagnosis of allergies.
Reviewer 3 Report
Comments and Suggestions for Authors
The study is well-designed and present in good shape. However, several key revisions are required to enhance clarity:
- Abstract: The abstract currently omits explicit mention of the key dust mite species identified and lacks quantitative results to provide concrete evidence of the observed patterns. Thus, limiting its impact and reader engagement.
- Linha 44 and 45: "genera Tyrophagus and Acarus".."Glyciphagus, Lepidoglyphus and Gohieria"..please, genera in italic;
Introduction:
The study's hypotheses are not explicitly stated at the end of the Introduction, making the research objectives and author expectations unclear. Add a concise, numbered list of hypotheses based on the study's rationale. Place this immediately before the aims paragraph.
Discussion:
The Discussion does not open with a clear assessment of which hypotheses were supported or refuted by the results. Begin the Discussion by directly addressing each hypothesis, stating whether it was fully corroborated, partially corroborated, or refuted, and briefly cite key supporting/contradicting results.
Methods:
The manuscript provides no explicit information regarding the exact number of mites per species used in the PCR experiments. The sole quantitative data related to molecular assays refer to sample mass (in mg) and extracted DNA quantity (in ng/μl), but not to the number of individuals per species. The critical question is: based on mite biomass and population sources, to what extent might this influence the results? Please, present these concerns and discuss their implications.
Age-stage distribution: larvae or adults were used like samples. It is unclear. This should be mentioned.
Author Response
Comments 1:
- Abstract: The abstract currently omits explicit mention of the key dust mite species identified and lacks quantitative results to provide concrete evidence of the observed patterns. Thus, limiting its impact and reader engagement.
Answer 1:
Already changed. Mentioned in the manuscript abstract
Comments 2:
- Linha 44 and 45: "genera Tyrophagus and Acarus".."Glyciphagus, Lepidoglyphus and Gohieria"..please, genera in italic;
Answer 2:
Already changed in the manuscript
Comments 3:
Introduction:
The study's hypotheses are not explicitly stated at the end of the Introduction, making the research objectives and author expectations unclear. Add a concise, numbered list of hypotheses based on the study's rationale. Place this immediately before the aims paragraph.
Answer 3:
Already added at the end of the introduction.
Comments 4:
Discussion:
The Discussion does not open with a clear assessment of which hypotheses were supported or refuted by the results. Begin the Discussion by directly addressing each hypothesis, stating whether it was fully corroborated, partially corroborated, or refuted, and briefly cite key supporting/contradicting results.
Answer 4:
Already added in discussion.
Comments 5:
Methods:
The manuscript provides no explicit information regarding the exact number of mites per species used in the PCR experiments. The sole quantitative data related to molecular assays refer to sample mass (in mg) and extracted DNA quantity (in ng/μl), but not to the number of individuals per species. The critical question is: based on mite biomass and population sources, to what extent might this influence the results? Please, present these concerns and discuss their implications.
Age-stage distribution: larvae or adults were used like samples. It is unclear. This should be mentioned.
Answer 5:
Already implemented in the manuscript
The cultures contained around 60-70 adult mites per milligram, as well as other unquantified larval stages.
Reviewer 4 Report
Comments and Suggestions for Authors
The paper presents a real time PCR-based method to distinguish between some of the species of mites most commonly involved in the sensitization of HDM allergic individuals. The rationale behind is that different species of mites contain similar but not identical allergens, and that the species discrimination could help the practitioner with the diagnosis and would allow a more taylored immunotherapy for the patient. Scientific evidences supporting this rationale are not properly provided by the authors. Moreover, some of the species of mites with the greatest differences in the allergen sequences, like Dermatophagoides and Blomia, thrive in very different geographic areas, therefore once a person has a suspect HDM allergy, only the extracts coming from the species most likely responsible for the sensitization should be used in principle. When this is not the case, and an "incorrect" diagnostic material or even allergenic preparation for immunotherapy is used, it is most likely because the "correct" one is not available, rather than because it is not known which kind of mite is responsible for the sensitization.
Although from the experimental results the selected primers seem to have proven their specificity, too few negative controls are included. Another point of concerns is why no Taqman probes have been used, that would have increased the specificity and even allowed to design a multiplexed version of the assay. The discussion paragraph is too concise and needs to be further expanded. Finally, the quality of the figures must be improved and some of the references are missing informations.
Author Response
The paper presents a real time PCR-based method to distinguish between some of the species of mites most commonly involved in the sensitization of HDM allergic individuals. The rationale behind is that different species of mites contain similar but not identical allergens, and that the species discrimination could help the practitioner with the diagnosis and would allow a more taylored immunotherapy for the patient. Scientific evidences supporting this rationale are not properly provided by the authors.
Answer:
The real objective of this study is to facilitate the identification of species present in patients' homes. The decision on the appropriateness of allergen-specific immunotherapy and its exact composition are the sole responsibility of the allergist and are based on European and American Guidelines.
Moreover, some of the species of mites with the greatest differences in the allergen sequences, like Dermatophagoides and Blomia, thrive in very different geographic areas, therefore once a person has a suspect HDM allergy, only the extracts coming from the species most likely responsible for the sensitization should be used in principle. When this is not the case, and an "incorrect" diagnostic material or even allergenic preparation for immunotherapy is used, it is most likely because the "correct" one is not available, rather than because it is not known which kind of mite is responsible for the sensitization.
The real objective of this study is to facilitate the identification of species present in patients' homes. The exact diagnosis of which mites the patient is sensitized to can only be carried out by in vivo testing (prick test) or in vitro tests (determination of specific IgE).
Comments 1:
Although from the experimental results the selected primers seem to have proven their specificity, too few negative controls are included.
Answer 1:
All necessary negative controls have been included in each PCR assay; as they are routine, they have not been included in the manuscript.
Comments 2:
Another point of concerns is why no Taqman probes have been used, that would have increased the specificity and even allowed to design a multiplexed version of the assay.
Answer 2:
We sought an easy, affordable, and universally accessible method. Taqman probes use is more expensive, and the probes are more elaborately designed, so they don't always work in multiplexing.
Comments 3:
The discussion paragraph is too concise and needs to be further expanded.
Answer 3:
Already implemented in the manuscript
Comments 4:
Finally, the quality of the figures must be improved and some of the references are missing informations.
Answer 4:
Already implemented in the manuscript

Round 2
Reviewer 1 Report
Comments and Suggestions for Authors
Dear authors,
I agree to the publication of the article. However, I recommend you read the text carefully and correct the missing letters etc (eg Figure 6: "unknown specie", similarly in the supplementary file).
Author Response
Comment 1
I agree to the publication of the article. However, I recommend you read the text carefully and correct the missing letters etc (eg Figure 6: "unknown specie", similarly in the supplementary file).
Answer 1:
Already reviewed and changed in the manuscript.
Reviewer 2 Report
Comments and Suggestions for Authors
The authors have done a great job of improving the manuscript, but there are still a number of comments that need to be taken into account.
- The manuscript requires extensive editing by a professional scientific editing service.
- The results section is overly descriptive and repetitive. The same conclusion (specific primers work, non-specific ones don't) is shown for every species individually, which could be condensed into a single, clear summary table and one representative figure. A summary table is needed early in the results, presenting for each species: amplicon size, melting Temperature (Tm), PCR efficiency, R² value, and the Ct value for self vs. non-self amplification.
- Combine figures 2, 3, and 4. A single, multi-panel figure showing amplification and melting curves for one species as a representative example is sufficient. The data for all other species should go into a Supplementary File.
- The claim that this method will be "highly useful for identifying allergy triggers in homes, hospitals, and schools" is an overreach based on the presented data. The study validates the method on pure and artificially contaminated cultures, not on complex, environmental dust samples. This is a critical limitation that should be acknowledged. Environmental samples contain PCR inhibitors, mixed DNA, and degraded material, which pose significant challenges not addressed here.
- The clinical utility is overstated. This statement needs to be tempered.
The English could be improved to more clearly express the research.
Author Response
Reviewer 2 – Round 2
Comment 1
The manuscript requires extensive editing by a professional scientific editing service.
Answer 1:
Figures have been edited and legends improved and sent in separate files. We have already asked for help to the IJMS Editorial Office.
Comment 2
The results section is overly descriptive and repetitive. The same conclusion (specific primers work, non-specific ones don't) is shown for every species individually, which could be condensed into a single, clear summary table and one representative figure. A summary table is needed early in the results, presenting for each species: amplicon size, melting Temperature (Tm), PCR efficiency, R² value, and the Ct value for self vs. non-self amplification.
Answer 2:
Table1 has been introduced before point 2.4., according to reviewer comment.
A summary table has been included early in the results, presenting for each species: amplicon size, melting Temperature (Tm), PCR efficiency, R² value, and the Ct value for self vs. non-self-amplification.
Comment 3
Combine figures 2, 3, and 4. A single, multi-panel figure showing amplification and melting curves for one species as a representative example is sufficient. The data for all other species should go into a Supplementary File.
Answer 3:
According to your comment, we have modified figures 2 and 3 of a representative example and included the rest of the data un combined the figures for one species and included the rest in a Supplementary File (figures 2s and 3s). Figures 4 and 5 have been removed (as they have been already shown).
Comment 4
The claim that this method will be "highly useful for identifying allergy triggers in homes, hospitals, and schools" is an overreach based on the presented data. The study validates the method on pure and artificially contaminated cultures, not on complex, environmental dust samples. This is a critical limitation that should be acknowledged. Environmental samples contain PCR inhibitors, mixed DNA, and degraded material, which pose significant challenges not addressed here.
Answer 4:
As described in section 2.5, the specificity of the oligonucleotides was analyzed in pure cultures of D. pteronnyssinus. These culture media use plant food and hydrolyzed yeast, which, combined with the presence of bacteria and fungi in the mites, makes the culture composition complex.
We have added in the Materials and Methods section that the cultures are prepared from plant food and hydrolysed yeast.
Comment 5
The clinical utility is overstated (exagerada). This statement needs to be tempered.
Answer 5:
Already reviewed and changed in the manuscript (mainly in Discussion)
Reviewer 3 Report
Comments and Suggestions for Authors
The revised version has been prepared in accordance with the comments. All suggested changes and recommendations have been carefully addressed.
Author Response
Reviewer 3 – Round 2
Comment 1
The manuscript is OK.
Answer 1:
Thank you very much for your revision.
Round 3
Reviewer 2 Report
Comments and Suggestions for Authors
The authors have made significant changes to the manuscript. However, some inaccuracies should be corrected before publication in IJMS.
- There are grammatical errors and typographical errors.
- The use of abbreviations is helpful but should be formally defined upon first use in the main text, not just in the Methods.
- Species names should be italicized consistently.
Author Response
Comment 1
There are grammatical errors and typographical errors.
Answer 1:
Already reviewed and changed in the manuscript.
Comment 2
The use of abbreviations is helpful but should be formally defined upon first use in the main text, not just in the Methods.
Answer 2:
Already reviewed and changed in the manuscript.
Comment 3
Species names should be italicized consistently.
Answer 3:
Already reviewed and changed in the manuscript.
Class, Subclass, Order, Suborder and Family are written in plain. Genus and species in italics.